# SPADE: spatial deconvolution for domain specific cell-type estimation
Yingying Lu[1], Qin M. Chen[2] & Lingling An [1,3,4] ✉

Understanding gene expression in different cell types within their spatial context is a key goal in genomics research. SPADE (SPAtial DEconvolution), our proposed method, addresses this by integrating spatial patterns into the analysis of cell type composition. This approach uses a combination of single-cell RNA sequencing, spatial transcriptomics, and histological data to accurately estimate the proportions of cell types in various locations. Our analyses of synthetic data have demonstrated SPADE's capability to discern cell type-specific spatial patterns effectively. When applied to real-life datasets, SPADE provides insights into cellular dynamics and the composition of tumor tissues. This enhances our comprehension of complex biological systems and aids in exploring cellular diversity. SPADE represents a significant advancement in deciphering spatial gene expression patterns, offering a powerful tool for the detailed investigation of cell types in spatial transcriptomics.

Spatial transcriptomics is a cutting-edge technology that has fundamentally transformed the field of transcriptomics by enabling studies of gene expression with unprecedented resolution and specificity[1]. The ability to identify the precise location of gene expression within a tissue represents a game-changing development, as it provides fresh avenues for investigating the complex interplay between gene expression and tissue architecture. By profiling the transcriptome at a high resolution in a spatial context, researchers can gain insights into the cellular heterogeneity that underlies normal tissue function or disease states[2], with significant implications for addressing a broad range of biological and medical questions. For example, spatial transcriptomics has demonstrated great promise in elucidating the cellular basis of brain function[3], and in enabling precision treatments for heart disease[4] or cancer[5]. Moreover, spatial transcriptomics has shown immense potential for studying the immune system[6]. Profiling the transcriptome of immune cells in various tissues has yielded insights into how the immune system responds to infection and disease[7]. This approach could play a crucial role in shaping the future of immunotherapies for cancer and other diseases, with spatial precision that is critical for effective treatment with minimal non-specific side effects[8].

Current spatial transcriptomics technologies face limitations in yielding cell type-specific information within a tissue region, thereby prohibiting the complete capture of gene expression patterns at single-cell resolution in space[9]. For instance, imaging-based spatial transcriptomics protocols provide detailed information at a single cell or subcellular level, but they are unable to measure a large number of genes, making them less suitable for exploratory investigations at the transcriptome level[10]. On the other hand, sequencing-based approaches allow for the measurement of gene expression for each spatial location across the entire transcriptome, but this comes at the cost of single-cell resolution[11]. As the compositions of cell types vary between different tissue locations, the data obtained from sequencing may be inconsistent for subsequent analyses. Specifically, when identifying differentially expressed genes across multiple spatial locations, the observed gene expression variations may not solely be influenced by spatial location, but also by differences in the categories or proportions of cell types[12]. Hence, there is a growing need for methodologies that accurately depict and describe the spatial patterns of gene expression variations while accounting for the specificity of individual cell types.

Single-cell RNA sequencing (scRNA-seq) has significantly advanced our understanding of cell heterogeneity and gene expression patterns at an individual cell level[13]. While scRNA-seq reveals intricate details of cellular functions, its limitation lies in not capturing the spatial context of cells within tissues[9]. Addressing this gap, computational deconvolution techniques have emerged, focusing particularly on integrating spatial transcriptomics with single-cell data. This integration is vital for understanding tissue architecture and the spatial distribution of cell types. Several spatially resolved cell type deconvolution techniques have been developed, including SPOTlight[14], spatialDWLS[15], RCTD[16], SpatialDecon[17], and CARD[12]. SPOTlight utilizes non-negative matrix factorization and non-negative least squares for cell type proportion calculation but neglects location correlations. RCTD leverages single-cell RNA-Seq data for cell type composition deconvolution while accounting for sequencing technology differences,

[1]Interdisciplinary Program in Statistics and Data Science, University of Arizona, Tucson, AZ 85721, USA. [2]College of Pharmacy, University of Arizona, Tucson, AZ 85721, USA. [3]Department of Biosystems Engineering, University of Arizona, Tucson, AZ 85721, USA. [4]Department of Epidemiology and Biostatistics, University of Arizona, Tucson, AZ 85721, USA. ✉e-mail: anling@arizona.edu

but it does not model spatial patterns. SpatialDWLS extends DWLS[18], employing a modified weighted least square for cell type composition estimation and uniquely using an enrichment test for cell type determination, but its enrichment score selection is arbitrary and spatial patterns are not considered. SpatialDecon surpasses traditional least-squares methods through log-normal regression and background modeling but overlooks location relationships. CARD incorporates conditional autoregressive modeling for spatial correlation structure consideration but disregards varying cell type identities in spatial patterns. Notably, none of these methods utilize valuable histological information. Overlooking spatial structures can lead to misleading conclusions, as they significantly impact biological functions.

To meet the challenge of cell type deconvolution in spatially resolved transcriptomics data, we have developed SPADE, a deconvolution tool that integrates cell type information derived from scRNA-seq data obtained from corresponding samples to accurately estimate the proportions of diverse cell types. Recognizing the unique characteristics of spatial transcriptomics data, such as the association of particular cell types with specific locations, the correlation between spatial positions and cell types, and the similarity between adjacent locations, we incorporated a cutting-edge spatial domain detection algorithm[19] that capitalizes on gene expression patterns, spatial coordinates, and histological data. To accommodate variations in cell type composition across distinct locations, we implemented an adaptive cell type selection step that efficiently determines the presence of specific cell types within each spot. Our findings substantiate the effectiveness of SPADE through rigorous simulations, wherein we benchmarked it against the existing spatial deconvolution methodologies. Furthermore, we applied SPADE to publicly available spatial transcriptomics studies across various areas, underscoring its utility in deciphering cell type-specific gene expression profiles. The proposed approach constitutes a significant advancement in the field of spatial transcriptomics, facilitating comprehensive and precise analyses of complex, heterogeneous tissue samples.

## Results

### Overview of SPADE

SPADE methodology involves a three-step approach to estimate the cell type proportions within a spatial domain, as depicted in Fig. 1 and Supplementary Fig. 1. In the first step, SPADE identifies the spatial domains within a tissue by employing spaGCN[19], a graph convolutional network specifically designed for spatial transcriptomics data. This integration of gene expression, spatial location, and histology data enables SPADE to identify the spatial domains that spatially coherent in both gene expression and histology. In the second phase, a cell type reference dataset is built from scRNA-seq to guide cell type identification within each domain, employing a Lasso regression algorithm[20]. This algorithm capitalizes on spatial gene expression data and cell type information to determine the optimal number of cell types present within each domain, which is subsequently employed for deconvolution analysis in the ensuing step. Concurrently, scRNA-seq data is adopted to create cell type-specific gene expression profiles, which guide the deconvolution process. In the final step, SPADE calculates the proportions of cell types within each spatial domain by utilizing cell type-specific features. These features consist of genes that are differentially expressed to each cell type. The SPADE analysis output provides the calculated cell type proportions for every spatial location for a given tissue region, which is an essential metric for investigating complex biological systems.

### Simulation studies

To simulate synthetic spatial gene expression data, we implemented a simulation approach similar to the CARD methodology[12], leveraging single-cell RNA-seq data. The synthetic data generation involved three steps: (1) generating random proportions for each spatial location within domains using a Dirichlet distribution and this proportion will be used as ground truth, (2) selecting cells from single-cell RNA-seq data within each cell type and summing these counts to produce cell type specific gene expression

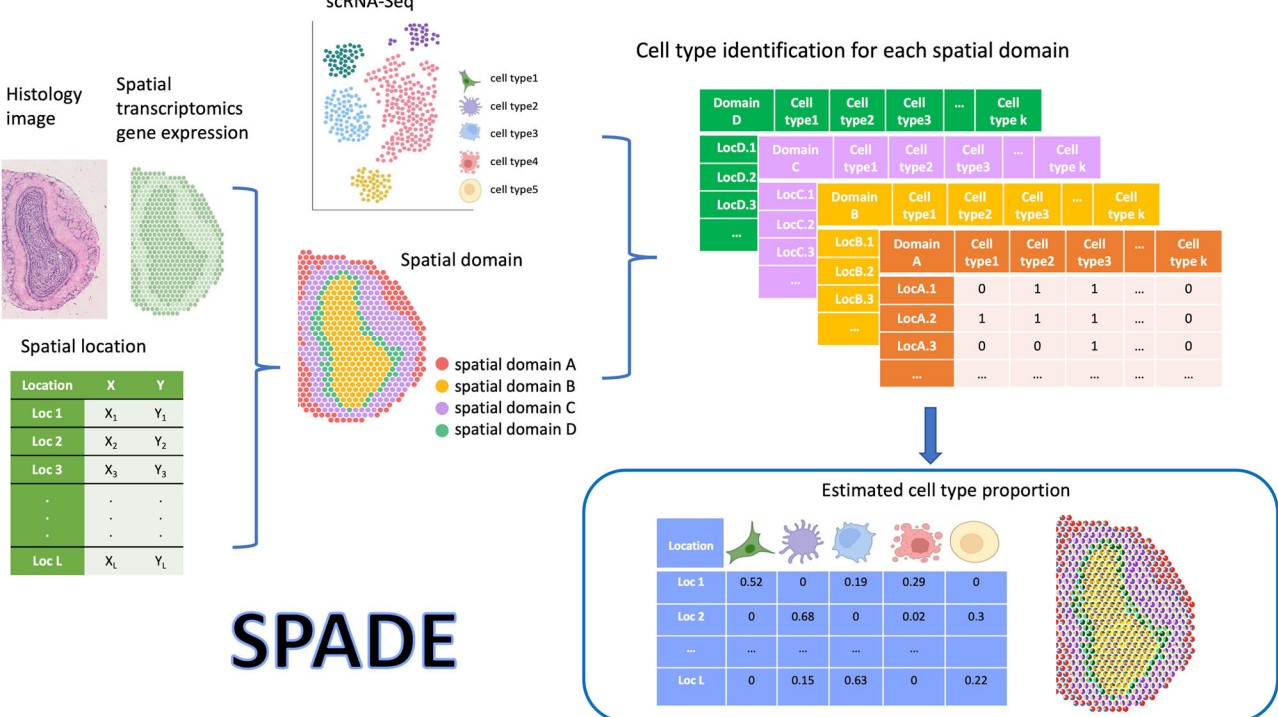

**Fig. 1 | Schematic overview of SPADE.** SPADE leverages reference single-cell RNA sequencing data to determine the cell type proportion at each location in the sample. To achieve this, SPADE first uses a combination of histology, spatial location, and gene expression information to identify spatial domains within a tissue. Subsequently, it performs a cell type selection for each domain by identifying the specific cell types present. Once the cell type information is obtained, SPADE utilizes scRNA-seq data to perform deconvolution, resulting in the estimation of cell type proportions for every spatial location. The final outcome of SPADE is the calculated cell type proportions for every spatial location. Part of this figure is created with BioRender.com.

data, and (3) aggregating gene expression across all cell types within each location and constructing a gene by location matrix as the pseudo-spatial transcriptomic data. More details can be found in Supplementary Fig. 2. This approach produced synthetic spatial gene expression data similar to real-world data. We conducted two separate simulation experiments that were generated on different mouse tissues. We also compared the method SPADE with existing spatial deconvolution methods, including CARD, SPOTlight, RCTD, spatialDWLS, and SpatialDecon.

The first simulation involves using mouse olfactory bulb (MOB) data. In this simulation study, we utilized three publicly available datasets to generate spatial transcriptomic data, including a single-cell RNA-seq dataset consisting of 10 cell types of the mouse olfactory bulb[21], a spatial gene expression dataset for the same area, and corresponding hematoxylin and eosin stain (H&E) image data (Fig. 2a)[22]. We employed SpaGCN and detected four distinct spatial domains (Fig. 2b), assigning a dominant cell type to each domain, along with varying numbers of minor cell types. To assess the accuracy of cell type detection, we created a bar plot displaying the true positive and false positive rates for each domain (Fig. 2c). This visualization highlights SPADE's capability to achieve the highest true positive rates and lowest false positive rates across all domains. The scatter plot (Fig. 2d) comparing the estimated and true proportions demonstrates that the SPADE estimation closely aligns with the ground truth, achieving results comparable to those of CARD. To represent the inferred cell type proportions for each spatial location, we employed a spatial scatter pie plot (Fig. 2e), in which SPADE generated an overall pattern that closely mirrored the true patterns and outperformed competing methods. Finally, to account for the stochastic nature of data generation, we evaluated SPADE and other methods by repeating the simulation ten times with varying proportions. The results are shown in a boxplot (Fig. 2f) in terms of mean absolute deviation (mAD), root mean squared error (RMSE), and correlation (R). Our results demonstrated that SPADE consistently outperformed other methods, achieving the lowest mAD and RMSE and the highest correlation across all simulations, followed by CARD.

To further investigate the performance of SPADE, we calculated the mAD and RMSE for each cell type, with the stacked bar plot signifying the least deviation in proportions inferred by SPADE (Supplementary Fig. 3a). Owing to the effective cell type selection of SPADE, its mean absolute deviation (Supplementary Fig. 3b) and correlation (Supplementary Fig. 3c) displayed superior outcomes across all cell types and domains in comparison to alternative methodologies. To visualize the estimation of dominant cell types, we employed a half violin plot (Supplementary Fig. 3d). This plot indicates that the distribution of dominant cell type proportions estimated by SPADE is more closely aligned with the true proportions than those obtained from other methods. We have also considered adding different levels of noise when generating the synthetic data, and compared the performance of SPADE with other methods on noisy data. From Supplementary Fig. 4, the results indicate that SPADE not only performs well under noisy conditions but also maintains its superior performance among all the compared methods.

In the second simulation study, we generated additional synthetic data from mouse kidney single-cell RNA-seq data[23] and obtained the mouse kidney spatial location and histology information from 10X Genomics to assess the robustness of our algorithm further. Specifically, we applied spaGCN and identified three spatial domains. SPADE was utilized to accurately retrieve the spatial pattern (Fig. 3a, b) compared to other methods (Supplementary Fig. 5) by assigning the most precise proportions to the dominant cell type within each spatial domain, as evidenced by Fig. 3c. To evaluate the accuracy of SPADE within each cell type across all locations, we compared the mAD and RMSE between true and inferred proportions to those obtained with other methods. Our analysis revealed that SPADE had the lowest error rate (Fig. 3d). Additionally, we created a scatter plot to compare the estimated cell-type proportions against the true proportions and found that SPADE displayed a close alignment to the 45-degree line (Fig. 3e). Furthermore, we assessed the ability of SPADE to accurately identify the correct cell types within each domain. Our analysis indicated the

superior ability of SPADE to detect the correct cell types in spatial locations, as evidenced by the high true positive rate and low false positive rate (Fig. 3f). Finally, we assessed the stability of SPADE's estimation by repeating the simulation ten times. The consistently low deviations, as well as high correlation (Fig. 3g), demonstrated that SPADE is a robust and accurate method for spatial deconvolution, superior to existing methods. To evaluate the ability of SPADE in handling noise data, we introduced varying levels of noise during the creation of synthetic data and compared its performance with other methods. The results, as shown in Supplementary Fig. 6, reveal that SPADE not only copes well with noisy conditions but also continues to maintain low deviance and high correlation among all compared methods.

## Application of real data on developmental chicken heart

The heart is the first organ to develop during embryogenesis, and interactions among various cell populations play a pivotal role in driving cardiac fate decision. The heterogeneity of cell types in heart development poses a challenge to study by traditional methods. Therefore, it is important to explore varied techniques for prediction of cell type heterogeneity during heart development.

During early embryonic development, the heart initially forms as a simple tube and undergoes a series of intricate morphological changes, eventually developing into a fully functional four-chambered heart complete with the blood vessels. In their previous research, Mantri, M. et al. employed a combination of spatially resolved RNA sequencing and high-throughput single-cell RNA sequencing to investigate the spatial and temporal interactions as well as the regulatory mechanisms involved in the development of the embryonic chicken heart[24]. Their research employed chicken embryos to generate over 22,000 single-cell transcriptomes across four pivotal developmental stages, in addition to spatially resolved RNA-seq on 12 heart tissue sections at the same stages, encompassing approximately 700 to nearly 2,000 tissue locations. These stages comprised day 4, an early stage of chamber formation and the initiation of ventricular septation; day 7, when the four-chamber cardiac morphology is initiated; day 10, representing the mid-stage of four-chambered heart development; and day 14, denoting the late stage of four-chamber development.

The study of early embryonic development details the progression of anatomical development across multiple temporal points by way of H&E stained images, as presented in the Supplementary Fig. 7a. Upon applying SPADE, spatial domains were defined for each timepoint, revealing the emergence of ventricular separation by day 4, as illustrated in Fig. 4a. From day 7 onwards, the clustering of diverse chambers was readily discernible, as evidenced by Fig. 4b, c, and d. The estimated cell type proportions for each chamber over the four temporal points are illustrated in the bar plot in Fig. 4e, which indicates a preponderance of immature myocardial and fibroblast cells on day 4, with a decreasing trend as the heart matures. This pattern is further verified by the scatter pie plot, as presented in the Supplementary Fig. 7b. This phenomenon is attributed to the tube-like structure of the chicken heart during early developmental stages, which necessitates the presence and active participation of fibroblast cells in the creation of connective tissue[25]. As the heart develops, fibroblast cells undergo proliferation and differentiation into various types of connective tissue cells. During later stages of development, the number of fibroblast cells in the heart declines, coinciding with its maturation and specialization. However, fibroblast cells continue to play a vital role in maintaining the heart's structure and function throughout the chicken's lifespan[26–29]. Conversely, the number of cardiomyocyte cells increases significantly during the development of the chicken heart, with the highest rate of proliferation occurring from day 4 to day 7, and slowing down from day 10 to day 14, as shown in Fig. 4e.

The proliferation of cardiomyocytes is a pivotal process during embryonic heart development, leading to a significant increase in their numbers. Previous studies have demonstrated that the rate of cardiomyocyte proliferation is highest during early developmental stages and gradually decreases as the heart matures[28,30–33]. Our findings, obtained through the application of SPADE, support this notion. Specifically, we observed that

immature myocardial cells constitute a subset of cardiomyocytes that are present only during days 4 to 7 of embryonic development (Fig. 4e). These immature cells undergo differentiation to become mature cardiomyocytes, which is a crucial step for the proper contractile function of the heart[24].

We aimed to investigate the trends in proportions of various cell types, including cardiomyocytes, vascular endothelial cells, fibroblasts, and endocardial cells. To determine whether the observed changes in proportions were statistically significant, we conducted a thorough analysis,

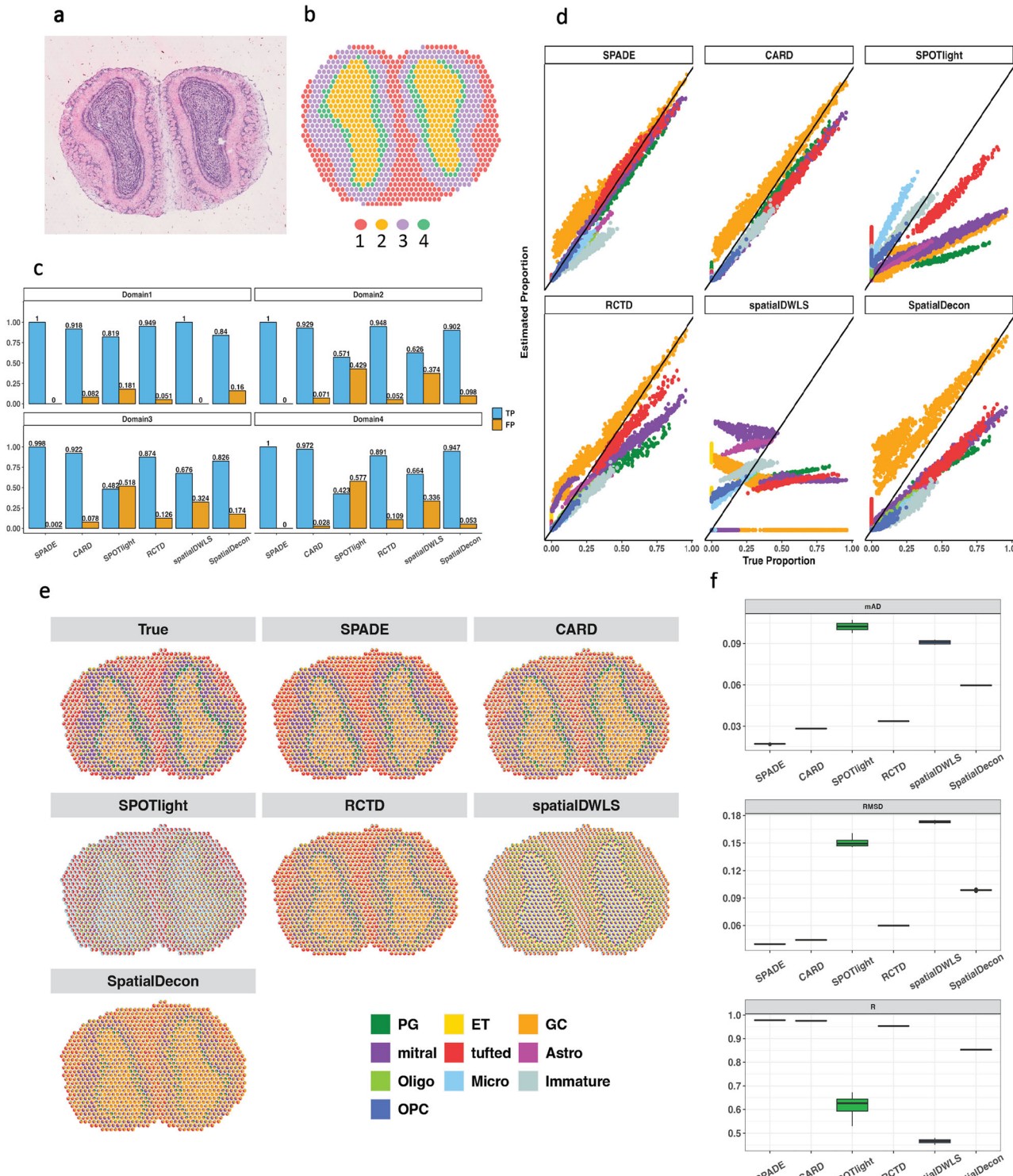

**Fig. 2 | Simulation using mouse olfactory bulb. a** H&E staining for the mouse olfactory bulb downloaded from[22]. **b** Spatial domain detection **c** True positive and false positive rate for detecting the correct cell types within each domain. **d** Scatter plot for comparing the estimated proportion with the true proportion. Each dot represents proportion at a location, with a color depicting a cell type. The color code is consistent with the color assigned in **e**. A 45-degree line indicates the same value for true and estimated proportion. **e** Spatial scatter pie plot shows the estimated cell-type composition on each spatial location from different deconvolution methods, compared to the true distribution. Colors represent cell types. **f** Boxplot of performance metrics for 10 simulation replicates. The overall simulation results indicate that SPADE outperformed other methods, achieving the lowest mean Absolute Deviation (mAD), Root Mean Square Error (RMSE), and the highest R. Source data can be found in Supplementary Data 1.

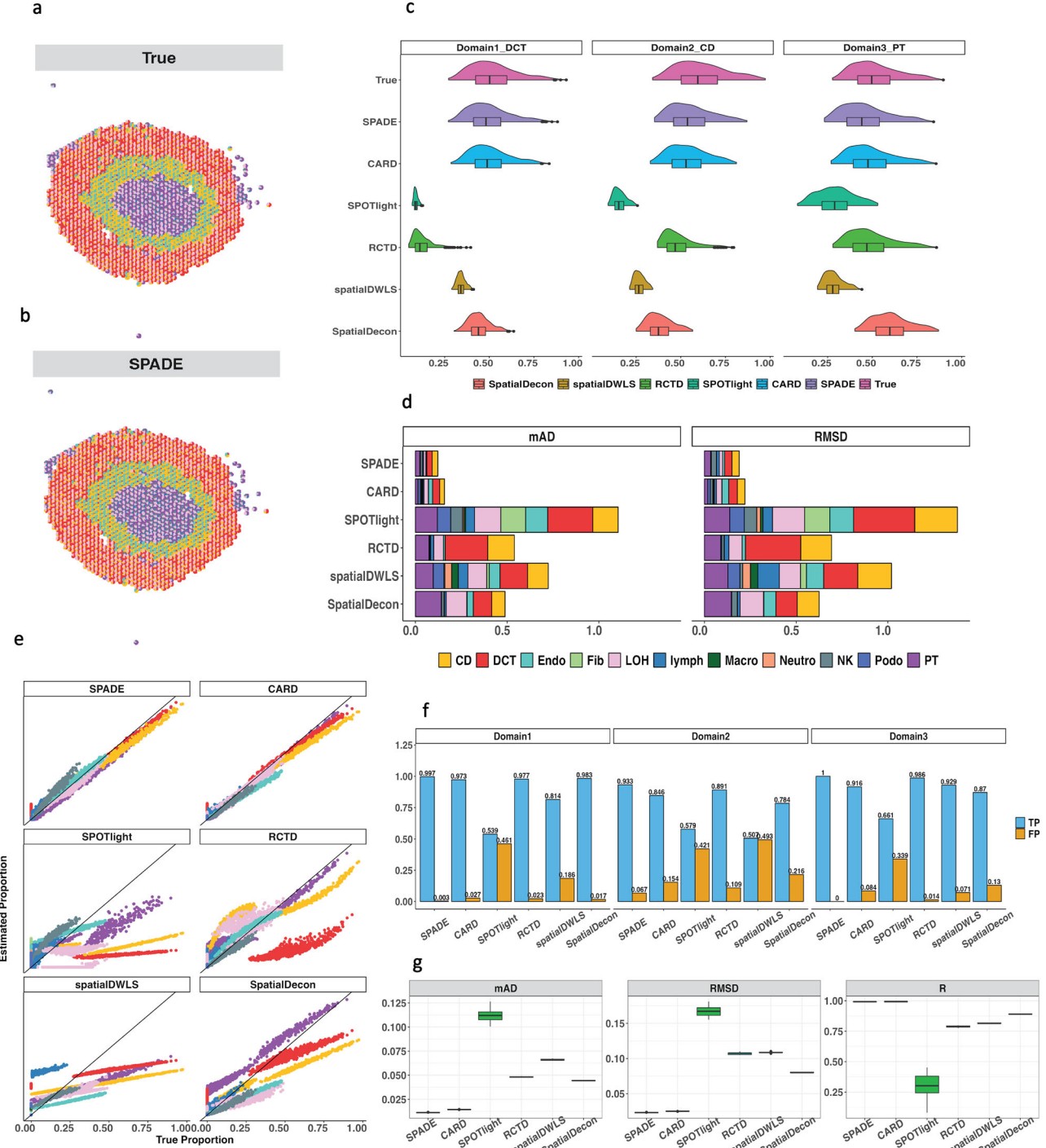

**Fig. 3 | Simulation using mouse kidney. a,b** Scatter pie plot representing cell type proportions within each location. Each location is depicted by a pie plot showing the cell type composition denoted by distinct colors. **c** Violin-box plot displaying the distribution of the predicted proportions of the dominant cell type within each domain compared with true proportion. **d** Stacked barplot exhibiting the mean absolute deviation and root mean square deviation between the true and predicted proportions. **e** Scatter plot showing cell type proportions, where each dot represents proportion at a location and the color corresponds to the cell type. **f** True positive and false positive rates for cell type identification within each domain. **g** Boxplot of performance metrics for 10 simulation replicates. Source data can be found in Supplementary Data 2.

comparing every pair of time points for each cell type using Fig. 4f. Our results indicate that nearly all changes between any two days were statistically significant (from Wilcoxon test with $p < 0.05$). Furthermore, we employed a spatial cell type map (Fig. 4g, h) to visually represent the proportions of each cell type at Day 4 and Day 14. Results for Day 7 and Day 10 can be found in Supplementary Fig. 8a. As expected, at Day 4, both cardiomyocytes and vascular endothelial cells exhibited relatively low

proportions in Fig. 4g, while at Day 14 (Fig. 4h), their proportions increased significantly. These findings highlight the dynamic changes in cell type proportions over time, providing crucial insights into the development and function of the studied tissues.

The heart, a vital organ composed of various cell types, including cardiomyocytes, fibroblasts, endothelial cells, and smooth muscle cells, undergoes intricate cellular interactions and network formation during its

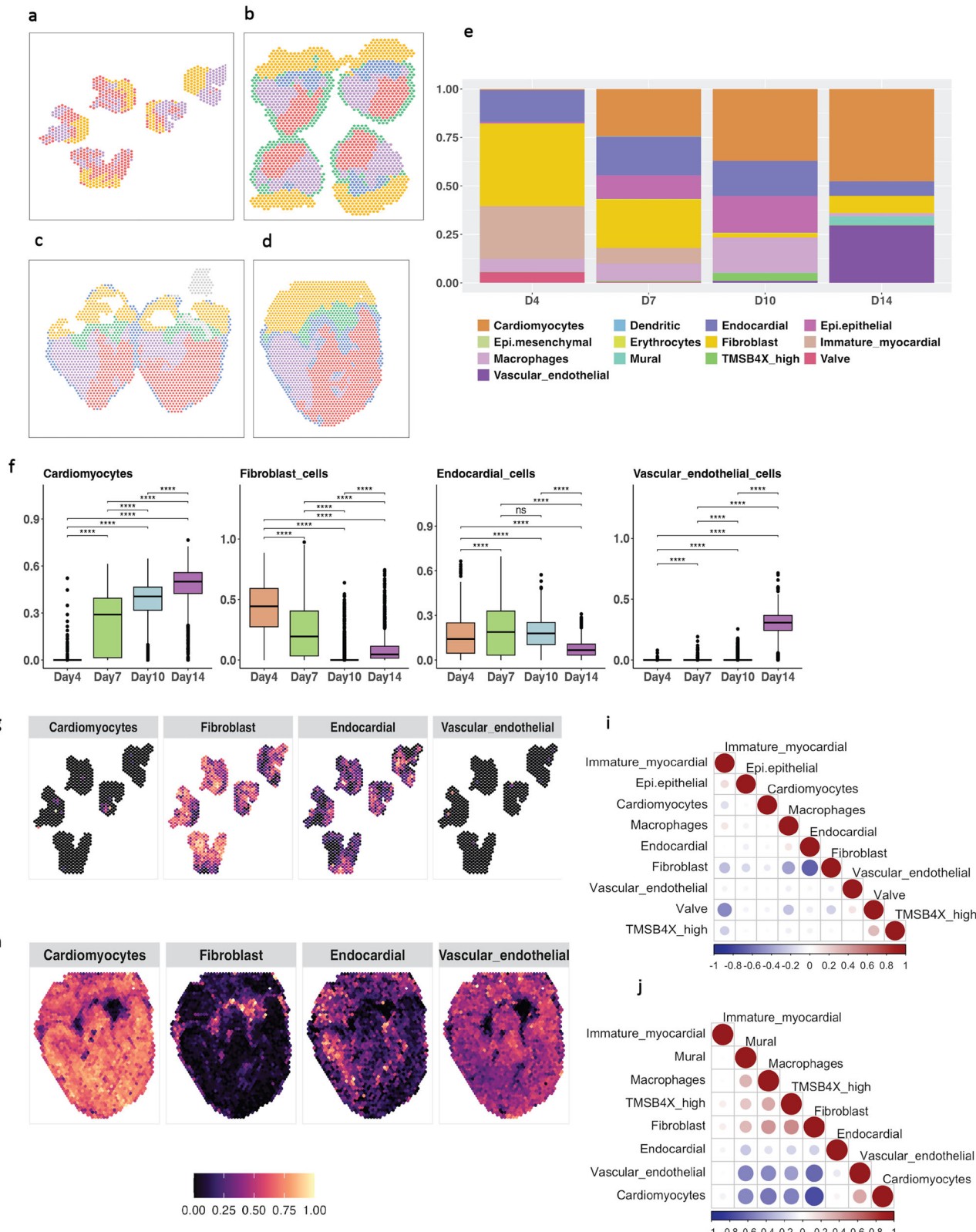

**Fig. 4 | SPADE application to developmental chicken heart. a–d** Estimated spatial domains for various time points during the experiment: (**a**) Day 4, (**b**) Day 7, (**c**) Day 10, and (**d**) Day 14. Colors indicate different domains, with an increasing number of domains detected as time progresses. Specifically, 3 domains were detected on Day 4, while 5 domains were identified on Day 7 and beyond. **e** Predicted cell type proportions during heart development, with colors representing different cell types. **f** Comparison of cell type proportions between time points, using a two-sided Wilcoxon Rank Sum test to assess differences for pairs of cell types. Asterisks indicate the significance level. **g, h** Scatter plots displaying the spatial locations of four selected cell types on Day 4 and Day 10, respectively, with each location colored according to the cell type proportion. **i, j** Correlation plots for cell type colocalization on Day 4 and Day 14, respectively. The size of the dot indicated the magnitude of the absolute correlations. Source data can be found in Supplementary Data 3.

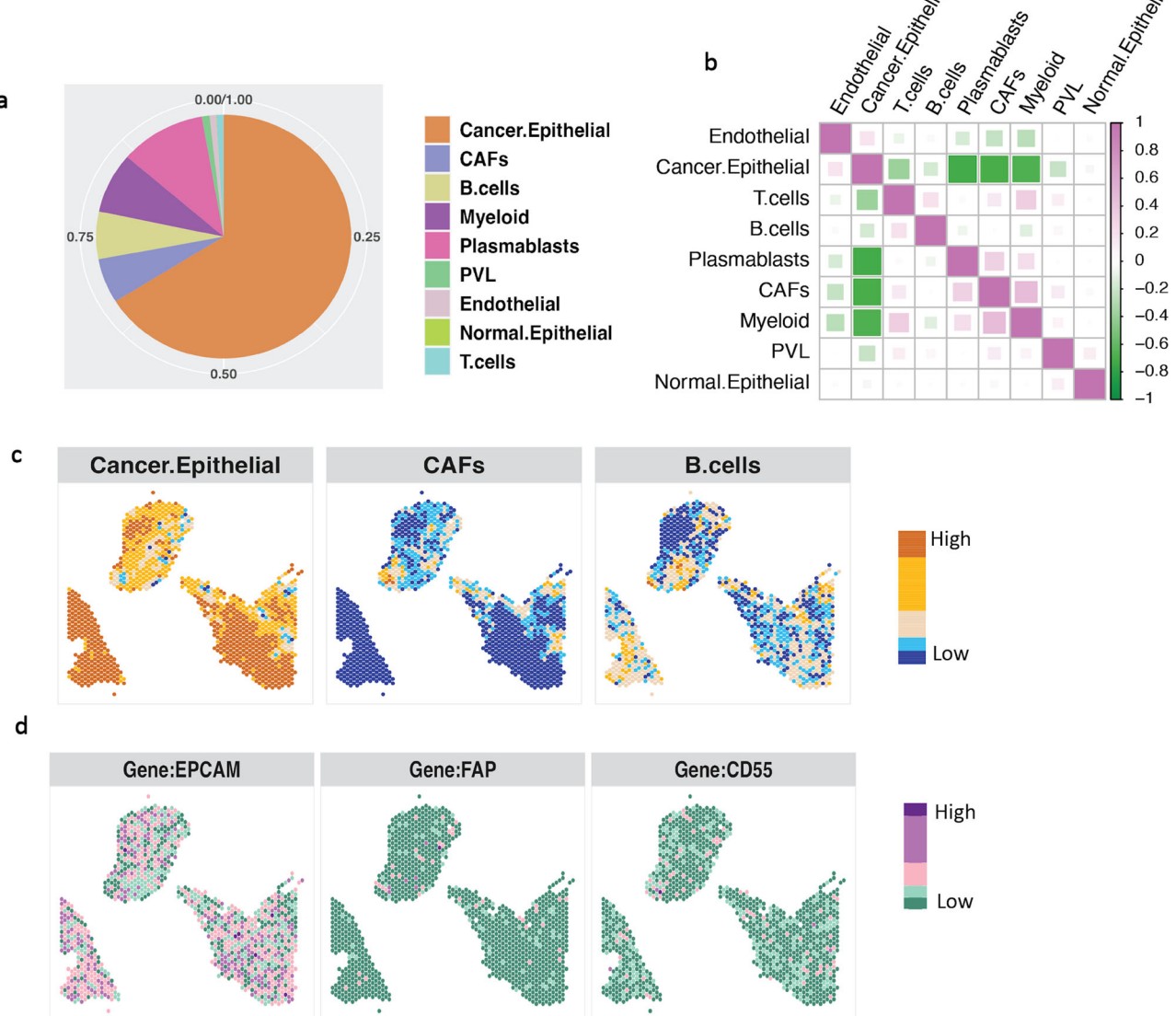

**Fig. 5 | SPADE application to human breast cancer. a** The estimated cell type proportions are shown, with different cell types represented by different colors. **b** Correlation for every pair of cell type proportions across the spatial location. **c** The cell type proportion for cancer epithelial, cancer-associated fibroblast (CAFs), and B cell is visualized in each location. **d** The marker gene expression levels for these three cell types are also displayed for each location respectively. Source data can be found in Supplementary Data 4.

developmental stages that are crucial for its proper functioning[34–37]. We utilized cellular colocalization analysis, a key technique in spatial transcriptomics, to quantitatively evaluate how different cell types are positioned and interact within tissue. This approach provides insights into the spatial dynamics of cellular environments, revealing potential interactions and functional relationships between cells. By analyzing the spatial organization and proximity of cell types, we aim to understand their roles in tissue function and development, and how they contribute to the overall tissue architecture and intercellular communication[35,36]. Our results revealed an increased cohesion between cell types, particularly between cardiomyocytes and vascular endothelial cells, in conjunction with heart development. This was supported by our results, as illustrated in Fig. 4i, j for Day 4 and Day 14, respectively, which showed stronger spatial coherence of organization during development. The correlation plots for Day 7 and Day 10 are in the Supplementary Fig. 8b. Collectively, our study highlights the significant variability in the spatial organization of cell types across different developmental stages and underscores the significance of dynamic interactions among various cell types for a comprehensive understanding of heart development as compared to the results from other methods (indicated in Supplementary Figs. 9–13).

## Application of real data on human breast cancer

Breast cancer is a complex disease that arises from the uncontrolled growth of malignant cells in the breast tissue, with varying molecular and cellular characteristics among individual patients. The Luminal subtype, which constitutes approximately 70% of all cases, is characterized by the expression of hormone receptors, namely estrogen receptor (ER) and progesterone receptor (PR)[38]. The combination of spatial transcriptomics and single-cell data is proving to be a valuable method for unraveling the complexities of human breast cancer[38]. This method maps gene expression and analyzes single-cell transcriptomes to identify cell types and their interactions in the tumor environment, crucial for understanding cancer progression and treatment effectiveness.

We retrieved the single-cell RNA-seq data as well as the spatial transcriptomics data of primary pre-treatment breast tumor samples from a human breast cancer study[39]. To create a reference for cell types in SPADE analysis, we utilized scRNA-Seq data comprising 9 distinct cell types from breast tumors. This reference was then employed to deconstruct a spatially mapped tumor sample. In the SPADE results (Fig. 5a), a preponderance of cancer epithelial cells is evident, with plasmablasts as the subsequent most

abundant cell type. A comprehensive examination of cellular composition across various spatial locations, depicted in Supplementary Fig. 14, further corroborates the prevalence of cancer epithelial cells at the majority of these sites. In Luminal breast cancer, the development of malignancy typically stems from epithelial cells, which may undergo genetic mutations leading to uncontrolled growth and tumor formation. These malignant epithelial cells often express high levels of hormone receptors, which facilitate response to the growth-promoting effects of estrogen and progesterone[40,41].

Plasmablasts are a type of immune cell that plays a crucial role in humoral immune response, responsible for antibody production and secretion. Recent evidence has shown that Luminal breast tumors with higher levels of infiltrating plasmablasts have a better prognosis compared to the tumors with lower levels of plasmablasts, suggesting a potential protective role of these cells in Luminal breast cancer[42,43]. We observed the colocalization of cancer epithelial cells and immune cells, such as plasmablasts, myeloid cells, and T/B cells, in the tumor microenvironment (Fig. 5b). We noted strong negative correlations between cancer epithelial cells and immune cells in these areas. The presence of tumor-infiltrating lymphocytes (TILs) is an important aspect of cancer epithelial cell and immune cell colocalization. TILs are immune cells that migrate into the tumor microenvironment and are believed to play a crucial role in anti-tumor immunity[42]. In several cancer types, including breast cancer, the presence of TILs has been linked to improved outcomes[43–45].Furthermore, We investigated the cell type proportion within each location for cancer epithelial cells, cancer-associated fibroblast cells (CAFs), and B cell (Fig. 5c), along with their associated marker genes EPCAM (Epithelial), FAP(CAFs) and CD55 (B cell) (Fig. 5d). The spatial distribution of cell types corresponded with their marker gene expression, confirming the cell types inferred by SPADE. The results displayed a similar pattern to those of CARD and RCTD, as shown in Supplementary Fig. 15.

### Application of real data on mouse visual cortex

The mouse brain, with its millions of neurons, is an ideal model for studying mammalian brain structure and function, especially in the visual cortex. This region, crucial for processing visual information, is organized into layers, each with specialized cell types, making it a good model for human visual cognition research[46–49]. The visual cortex hosts various neuron types, including excitatory neurons using glutamate and inhibitory neurons using GABA, forming a network for interpreting visual stimuli[50,51]. Each neuron type plays a specific role in visual processing, from detecting visual features to integrating complex visual information[52–54]. Understanding these functions and their disruptions can provide insights into neurological and psychiatric disorders[55]

We implemented a single-cell analysis[56] to identify 30 distinct cell types in the mouse visual cortex. This analysis was used to deconstruct the adult mouse brain, which had undergone spatial processing (see Fig. 6a). Initially, we divided the mouse brain into 19 different regions (illustrated in Fig. 6b). In these regions, we were able to identify specific layers that correlate with various brain functions. Compared to the other methods (results are in Supplementary Fig. 16), SPADE successfully decomposed each brain region into its constituent cell types. The predominant cell type in each location is shown in Fig. 6c. Our focus was particularly on the visual cortex, where we found that most areas were primarily composed of excitatory neurons, followed by inhibitory neurons and oligodendrocytes, as detailed in Fig. 6d. Excitatory neurons, which utilize the neurotransmitter glutamate to typically enhance neuronal activity, are an integral part of the mouse visual cortex, as well as all mammalian brains. These neurons have a central role in transmitting and processing visual data. They are found in all layers of the mouse visual cortex from the deeper layers (layers 5 and 6) to the superficial layers (layers 2 and 3) (Fig. 6e). The genes expressed differently (Fig. 6f) for each subtypes of excitatory neuron further confirmed the corresponding multiple-layer structures.

### Discussion

Spatial transcriptomics, essential for studying gene expression and tissue diversity, is more informative when combined with cell type deconvolution.

This computational method identifies cell types from gene expression data, enhancing our understanding of biological processes at the cellular level within tissues. Spatial transcriptomics is a critical tool for investigating gene expression patterns and regional differences in a tissue, providing insight into its biological significance. However, the interpretation of this data can be challenging without knowledge of the specific cell types present in each region. Cell type deconvolution is a computational approach that can identify cell types based on gene expression data. By applying this technique to spatial transcriptomics data, it becomes possible to contextualize gene expression data and gain a deeper understanding of the biological processes occurring within a tissue at the cellular level. While many existing cell type deconvolution methods do not account for the spatial domain structure, SPADE has been developed to overcome this limitation. Our method stands out by integrating spatial structures and using a reliable approach for cell type selection. Differing from other techniques, it employs lasso regression and adaptive thresholding for more accurate and flexible cell type identification. This effectiveness is evident in our results, notably in Fig. 2c, enhancing SPADE's robustness and precision in complex spatial transcriptomics datasets.

The SPADE algorithm effectively predicts cell types and their distribution in tissues, as shown in tests on synthetic mouse datasets. Applied to chicken heart development, human breast cancer, and mouse visual cortex, SPADE revealed insights into cell type development and spatial patterns in diseases. This has promising implications for clinical studies, especially in understanding cancer cell type heterogeneity and informing treatment strategies.

Although the SPADE algorithm has demonstrated superior accuracy, one of its notable challenges is the accurate deconvolution of rare cell types. Our analysis, especially with rare cell types like the Immature cells in the mouse olfactory bulb data, indicated a tendency for underestimation, a limitation common to current deconvolution methods. This underestimation issue is critical to address in order to improve SPADE's robustness and applicability, particularly in complex biological tissues where rare cell types are are crucial for functional significance or disease state.

Moreover, it's important to note that SPADE's performance can be further improved by incorporating better-designed reference datasets. In our study, we utilized one single scRNA-seq dataset to construct the reference, potentially limiting the algorithm's overall efficacy. Advances in scRNA-seq technologies have led to the generation of multiple reference datasets from different platforms or samples obtained from the same tissues. Integration of these diverse scRNA-seq datasets holds the potential to provide a comprehensive and accurate reference set, thereby improving the performance of the SPADE algorithm.

It should also be mentioned that SPADE, while not the most efficient in processing time and memory usage, involves a meticulous process of identifying cell types within each domain before estimating proportions. This methodological aspect, though extending processing time, substantially enhances the accuracy and robustness of our analyses, particularly for complex datasets. This balance between processing efficiency and analytical precision is a key consideration, making SPADE a valuable tool for in-depth spatial gene expression studies. Continuous methodological improvements are necessary. Future studies should explore the use of multiple reference datasets to improve the accuracy and efficacy of SPADE in predicting cell types and their spatial distribution across different tissues.

### Methods

#### Spatial domain detection

Spatial domain detection constitutes a critical aspect of spatial transcriptomics, as evidenced by numerous studies[19,57,58]. A spatial domain encompasses regions that demonstrate spatial coherence in both gene expression and histology. Traditional approaches for identifying these domains are dependent on clustering algorithms that solely consider gene expression, neglecting spatial information and histology[19]. To address this limitation, spaGCN[19] incorporates gene expression, spatial location, and

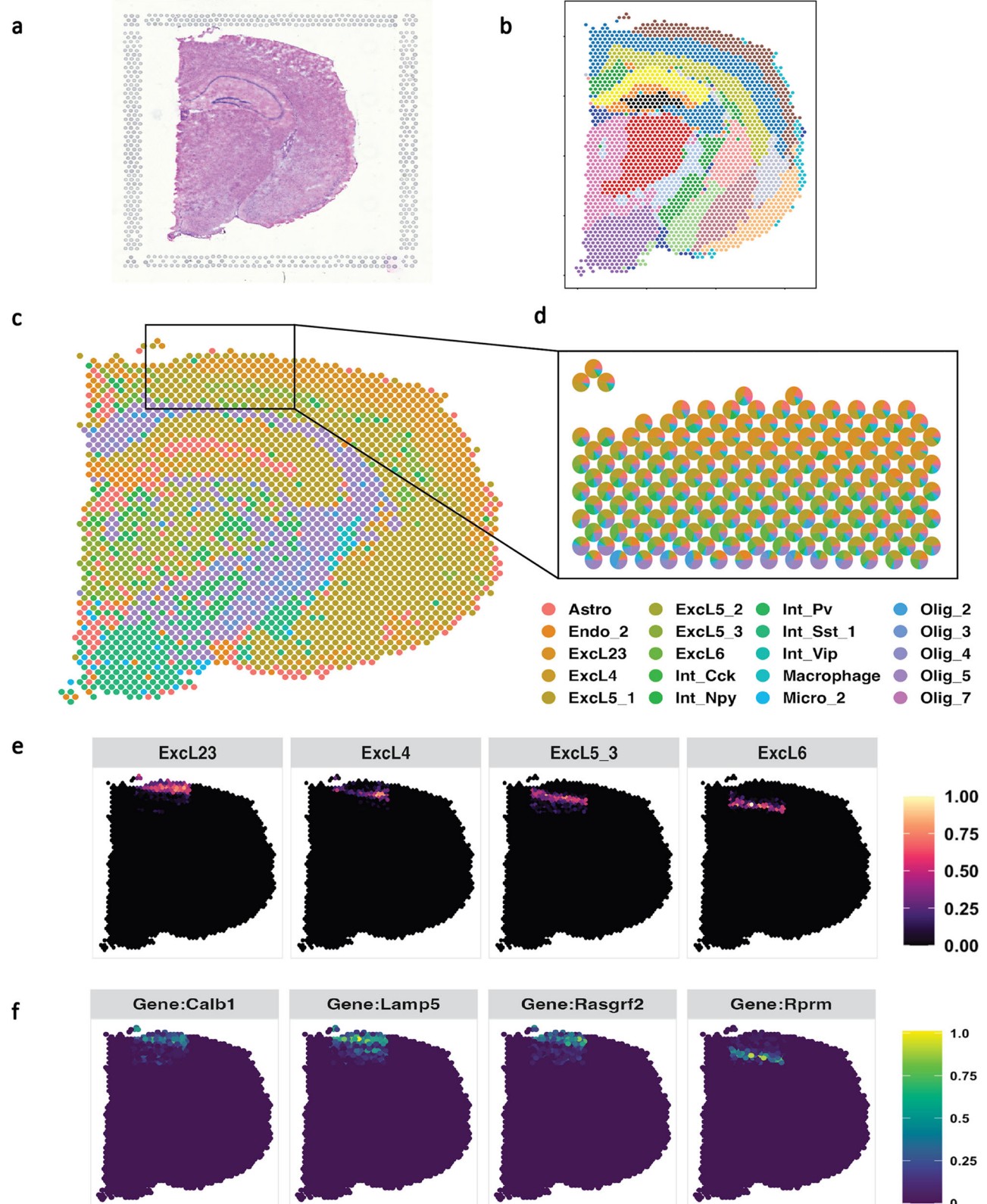

**Fig. 6 | SPADE application to Mouse Visual Cortex. a** Original image of Adult Mouse Brain (Coronal) downloaded from 10x Visium. **b** Detected spatial domain. Colors represent different domains **c** SPADE inferred the dominant cell type at each location. **d** Estimated cell type in the mouse visual cortex. Each location is indicated by a composition of several cell types. **e** 4 subtypes of the excitatory neurons at the mouse visual cortex. **f** Genes displayed differences in expression within each excitatory neuronal cell subtype at the mouse visual cortex. Source data can be found in Supplementary Data 5.

histology to construct a graph convolutional network, facilitating the identification of spatial domains. The spaGCN algorithm unfolds in three stages. Initially, information derived from physical location and histology is employed to establish an undirected graph, reflecting the relationships between all spots. Subsequently, a graph convolutional network is implemented to integrate gene expression, spatial location and histological data. Finally, an iterative unsupervised clustering algorithm is applied to segregate spots into distinct spatial domains based on gene expression and histology coherence. Importantly, spaGCN can be applied to the datasets where histology images are absent. In these situations, it makes use of spatial gene expression data to identify the spatial domains which is comparable to methods used in other spatial domain detection approaches. For a comprehensive understanding, refer to the original publication[19].

### Determine the number of cell types for each domain

A crucial disparity between bulk deconvolution and spatial deconvolution is that not all cell types are uniformly distributed across all regions. Consequently, identifying the presence of specific cell types in individual locations is crucial for efficacious cell type deconvolution. A key assumption underlying this approach is that while different locations within the same domain are closely related, they may not have exactly the same cell types. Instead, each location is thought to contain a similar set of cell types, but the proportions of these cell types can vary from one location to another. To tackle this issue, we leverage a Lasso-regularized generalized linear model[20], which offers the advantages of concurrent feature selection and regularization, enforcement of sparsity, computational efficiency, resistance to multicollinearity, and broad applicability across diverse domains. Employing Lasso, cell types are selected for each domain through the subsequent methodology:

$$\sum_{i=1}^{M}\left(y_i - \sum_{j=1}^{K}\beta_j x_{ij}\right)^2 + \lambda\sum_{j=1}^{K}|\beta_j| \qquad (1)$$

where $y_i$ is the gene expression for gene $i$, $x_{ij}$ is the gene $i$ expression for cell type $j$, $\beta_j$ is the coefficient for cell type $j$. To perform cell type selection, we estimate cell type coefficients, effectively eliminating a cell type from a given location if its coefficient shrinks to 0. The tuning parameter, $\lambda$, is chosen via 10-fold cross-validation.

Upon obtaining the cell-type-associated coefficient matrix for each location within the spatial domains, we transform it into a binary matrix, where each entry holds a value of either 1 or 0. To achieve this, we employ an adaptive thresholding technique[59] that utilizes a 2D convolution with the Fast Fourier Transform (FFT) to filter the coefficient matrix, thus enabling the efficient identification of entries surpassing a specific threshold. In particular, if a coefficient exceeds the filtered value, the corresponding entry is set to 1, whereas entries falling below the threshold are assigned a value of 0. A comprehensive description of these steps can be found in Supplementary Fig. 17.

### Cell type proportion estimation for each location within each domain

The deconvolution problem can be solved to find the optimal estimation for cell type proportion that minimize the difference between estimated spatial gene expression and observed spatial gene expression for each location as below:

$$\underset{P}{\mathrm{argmin}}\left\{\sum_{i=1}^{M}\left|y_i - \sum_{j\in S}p_j x_{ij}\right|\right\} \qquad (2)$$

subject to $p_j \geq 0$ and $\sum_{j\in S}p_j = 1$ where $y_i$ is the expression for gene $i(=1\ldots M)$. $x_{ij}$ is the expression for gene $i$ for cell type $j$ that is extracted from single-cell reference. $p_j$ is the proportion for cell type $j$. $S$ is the set of cell types determined for each domain. Here we select the absolute deviation loss as the optimal choice due to its less sensitive to the extreme values than

the commonly used quadratic loss function. The optimization problem is solved using the Augmented Lagrange Minimization algorithm that is implemented by auglag function in R package alabama[60]. Due to the unique feature of proportion, we not only minimize the nonlinear objective function, but also satisfy two constrains. The proportion for each cell type has to be nonnegative, and the sum of all cell type within each sample needs to be 1.

### Construct reference

The accurate estimation of cell types is essential for understanding tissue function and identifying cell type specific features. A well-designed cell type reference is crucial for this purpose, and in this study, we utilize single-cell RNA-seq data that contains tissue or samples with a similar phenotype to the spatial transcriptomics data. The scRNA-seq data were first checked for quality based on the commonly used pre-processing workflow from Seurat[61].

To extract cell type information, we followed the main idea from MuSiC[62] and applied several steps. Firstly, we calculated the cross-cell variation for each gene of each cell type within an individual sample, taking into account cell type and sample-specific library size. To achieve this, we subset the expression data by removing redundant cell type annotations given by the original single-cell study and by removing genes with zero counts. For each sample within each cell type, we scaled the gene expression by their library size, which is calculated by summing all gene counts for each cell. Next, we filtered genes by three criteria to keep genes that satisfy any of these criteria: 1) the genes shared between single-cell data and bulk data, 2) commonly used cell type biomarkers or highly cited markers, and 3) differentially expressed genes (DEGs) by comparing each pair of cell types. To detect DEGs, we used the FindAllMarkers function from Seurat. The resulting table is a gene by cell type expression matrix that can be implemented in the cell type deconvolution model. For a more in-depth step of reference construction, please refer to the flowchart depicted in the Supplementary Fig. 18 for more details.

### Statistics and reproducibility

All single-cell and spatial transcriptomics data used for simulation and real datasets are publicly available. The codes for other methods are also publicly accessible; we adhered to the online tutorials for running each method. For our method, we have developed an R package that enables the reproduction of our results. The R package and tutorial for implement SPADE is freely available on GitHub (https://github.com/anlingUA/SPADE).

### Reporting summary

Further information on research design is available in the Nature Portfolio Reporting Summary linked to this article.

### Data availability

The datasets utilized in this study are publicly available. The spatial MOB, mouse kidney, and mouse brain datasets were obtained from the 10x Visium dataset, which can be accessed at https://www.10xgenomics.com/resources/datasets. The single cell RNA-seq data for the MOB and mouse kidney samples are available through the GEO Series accession numbers GSE162654 and GSE107585, respectively. The spatial transcriptomics data for the developmental chicken heart were downloaded from https://github.com/madhavmantri/chicken_heart/tree/master/data, while the corresponding single cell data can be accessed via the GEO Series accession number GSE149457. The human breast cancer spatial transcriptomic data is available from the Zenodo data repository (https://doi.org/10.5281/zenodo.4739739), and the single cell data can be obtained via the GEO Series accession number GSE176078. Finally, the single cell data for the mouse visual cortex can be accessed through GSE102827.

**Article**

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

## Acknowledgements
This research was partially supported by the National Institute of Health R01 GM125212, R01 GM126165, and Holsclaw endowment (Q.M.C.); R01 GM139829, P01 AI148104-01A1, and United States Department of Agriculture (ARZT-1361620-H22-149) (L.A.).

## Author contributions
Conceptualization, L.A. and Y.L.; methodology, Y.L. and L.A.; simulation studies, Y.L. and L.A.; real data analysis, Y.L., Q.M.C., and L.A.; writing and revising the manuscript, Y.L., Q.M.C. and L.A.

## Competing interests
The authors declare no competing interests.
