## [Peer Review File · Communications Biology]

Reviewers' comments:

Reviewer #1 (Remarks to the Author):

Thank you for your interesting and complex work presented in "SPADE: Spatial Deconvolution for Domain Specific Cell-type Estimation". While the subject is fascinating and of great significance, there are a few areas where further clarity would be beneficial for readers:

1. The manuscript refers to the estimation of cell type proportions at "every spatial location". For greater clarity, it would be helpful to specify what is meant by 'every spatial location'. Are we discussing specific locations of all possible locations within a certain area? The significance of location could be better explained, as it is not immediately clear if the focus is on different cell types at a specific tissue location or each location having a specific cell type. If the method does what I understand it to do, estimating cell proportions at each location given cell type names and cell locations, then could you elaborate on the novelty of this method?

2. The manuscript mentions the construction of a "cell type reference dataset from scRNA-seq". Could you clarify how this is built? Does SPADE require pre-information of cell type annotation to calculate cell type proportion? In Figure 1, does the black color Spade like figure represent the SPADE algorithm's logo? Adding proper legends and annotations to the figures would aid understanding.

3. The method section discusses three stages: Spatial domain detection, determining the number of cell types for each domain, and cell type proportion estimation for each location within each domain. An overview figure depicting these steps, inputs, and outputs would provide a better understanding. It would also be interesting to see a performance evaluation of each of these stages, both individually and in combination.

4. While you discussed results and performance on simulated data, inclusion of results on real data would be beneficial for readers.

5. The robustness of SPADE in the face of noise would be of interest. Perhaps the method's performance could be discussed in relation to at least two datasets?

6. In Figure 4, you discuss the significance of changing cell proportions over time. Could you elaborate on the importance of these changes? Are they indicative of cell transition? If so, some analysis or evidence to support this would be beneficial.

7. Introduction and explanation of "colocalization analysis" and its importance would be beneficial.

8. Improvement could be made to all sections, specifically the Abstract and Results sections, to enhance reader comprehension.

Your research is indeed compelling, and with these additional details, it promises to be even more insightful and accessible to readers.

Reviewer #2 (Remarks to the Author):

Overall the concept and the work seemed both important and nice to me. And this will be very important to reveal the disease biology of different tissue. The writing language is very lucid, easy to understand and conceptualize.

Major comments:

1. In the stage of applying the method in real-world data, it is unclear if histology data is required. Within the "Application on Real Data" section the authors used histology data for the first and the third experiment, but didn't mention histology data on the second experiment where they applied SPADE to the breast cancer dataset.
2. It will be helpful to know if this method can be applied to spatial datasets where instead of a whole transcriptome, a panel of genes were used.
3. It is evident that single cell (reference) data set will not be from the same patient and the same region from where the spatial dataset will be collected. Thus it will be important to know how this tool will handle conditions where there are some domains containing some additional cell types or states, not identified in single cell (reference) dataset. This issue can be addressed by a simulation approach.
4. How does the presence of rare cell types affect deconvolution?
5. When simulating data for cell type selection, how is care taken to avoid bias which might result from cells with very extreme expression profiles or cells over-represented in the data?
6. The results from other deconvolution methods applied to real datasets need to be included and compared with the SPADE results.
6. How does SPADE take care of spatial noise and batch effects that might be present in spatial transcriptomics data?

Minor Comments:

Introduction:

1. The introduction part can be made little brief by minimizing statements on scRNA-seq and deconvolution of scRNA-seq and more focusing on spatial deconvolution.
2. It would be better if one statement is there about time and memory effectiveness of SPADE over other spatial deconvolution methods available under comparable system configurations. That would be more attractive to the users!
3. I am really eager to understand how SPADE will work if histology images are not available for some spatial datasets? How it will identify spatial domains in these cases?

Results:

1. Figure - 1: The layout is completely fine. Just a technical error is that the 'L' of locA.3 in Domain A of the figure is in lower case while all other is in upper case.

2. Figure - 2: In text explanation (mouse olfactory bulb), it would be better if one statement is added with respect to significance of SPADE having least mAD, RMSE and highest R across simulations.

Discussion:

1. It should include all strengths of SPADE for example along with spatial domain, any other advantages over other methods.
2. Also if there are some significant differences between synthetic or simulated datasets vs. real-world datasets on the performance of SPADE.

We thank the reviewers for insightful comments. The manuscript has been revised to incorporate these comments and highlighted in red. Our point-to-point response to each comment is shown below, and we also included the revised text within the reply.

Reviewers' comments:

Reviewer #1 (Remarks to the Author):

Thank you for your interesting and complex work presented in "SPADE: Spatial Deconvolution for Domain Specific Cell-type Estimation". While the subject is fascinating and of great significance, there are a few areas where further clarity would be beneficial for readers:

1. The manuscript refers to the estimation of cell type proportions at "every spatial location". For greater clarity, it would be helpful to specify what is meant by 'every spatial location'. Are we discussing specific locations of all possible locations within a certain area? The significance of location could be better explained, as it is not immediately clear if the focus is on different cell types at a specific tissue location or each location having a specific cell type. If the method does what I understand it to do, estimating cell proportions at each location given cell type names and cell locations, then could you elaborate on the novelty of this method?

"every spatial location" refers to all potential locations within a specified tissue region. This clarification aligns with the limitations of current technology highlighted in our introduction. For instance, the technology of 10x Genomics, a leading spatial genomic sequencing method, yields an average of gene expression for each location containing multiple cells, ignoring cell type distribution. Therefore, it becomes necessary to accurately decompose cell types at each specific location.

The novelty of our method lies in its capacity to estimate cell type proportions at each location (e.g., each location may contain 10 or 20 cells). Unlike most spatial deconvolution methods, our technique not only accounts for the possibility of different cell types present at various locations but also integrates histological information that reflects the underlying tissue structure. This integration of tissue structure in the analysis allows for a refined and location-specific understanding of cell type distribution, which distinguishes our method from others in the field of spatial deconvolution.

2. The manuscript mentions the construction of a "cell type reference dataset from scRNA-seq". Could you clarify how this is built? Does SPADE require pre-information of cell type annotation to calculate cell type proportion? In Figure 1, does the black color Spade like figure represent the SPADE algorithm's logo? Adding proper legends and annotations to the figures would aid understanding.

The detailed steps for constructing the reference dataset are described in the supplementary file, with figure illustrations to enhance clarity. Briefly, the process involves:

- 1) Calculating the average gene expression for each cell type within each donor or tissue sample.

- 2) Scaling this expression by a factor calculated based on the number of cells for each cell type and sample.
- 3) Averaging these scaling factors across all donor or tissue samples.
- 4) Applying these factors to scale the cell type-specific mean gene expression.
- 5) Finally, averaging this over all samples to obtain a sample and cell type-specific gene expression.

This refined gene expression data serves as a reference for inferring cell type information during cell type deconvolution.

Regarding the SPADE algorithm, it indeed requires pre-annotated cell type information as input, which is a standard feature for reference-based methods in cell type deconvolution. The information for this input is derived from the reference dataset.

Lastly, the black color spade-like figure in Figure 1 is not the SPADE algorithm's logo; it is a graphical representation per the algorithm's name. In the revised version, we have removed this to avoid confusion. The schematic overview of SPADE has been updated (Figure 1).

3. The method section discusses three stages: Spatial domain detection, determining the number of cell types for each domain, and cell type proportion estimation for each location within each domain. An overview figure depicting these steps, inputs, and outputs would provide a better understanding. It would also be interesting to see a performance evaluation of each of these stages, both individually and in combination.

We understand the point about the importance of a clear and concise overview figure to outline each stage of our method. To incorporate this suggestion, we have developed a workflow that provides a quick reference to the method stages along with their respective inputs and outputs.

This diagram has been designed to offer a high-level overview that complements the detailed figures for each specific step, which have been included in the manuscript (Figure 1). In the revised version, we have included this new workflow diagram in the supplementary file as shown below.

Green square shapes represent data inputs and outputs, while orange diamond shapes denote the stages or steps of the analysis. The workflow begins with data inputs, which include a Histology Image, Spatial Transcriptomics Gene Expression Data, and Spatial Location information. These inputs feed into the first stage, “Spatial Domain Detection”, where spatial domains are identified for each location. The output from this stage is “Spatial Domain for Each Location”, indicating the categorization of spatial locations into domains. Concurrently, “Reference Construction” is a stage that uses scRNA-seq data to create a cell type reference. The cell type reference, an output depicted as “Cell Type Reference from scRNA-seq”, is then utilized in the “Cell Types Determination for Each Domain” stage, contributing to the results as “Identified Cell Types For Each Spatial Domain”. The final stage, “Cell Type Proportion Estimation”, approximate the proportions of cell types across locations, culminating in the output “Cell Type Proportions for Each Location within Each Domain”, which represents the distribution of cell types within the spatial framework of the domains.

Regarding the performance evaluation of each stage:

- 1) Spatial Domain Detection: This stage utilizes an existing method, as mentioned in our manuscript. We have not conducted a separate performance evaluation for this stage since the original publication of the method has already established its efficacy. Our application of this algorithm integrates histology data, aligning with the existing validation.

- 2) **Determining the Number of Cell Types for Each Domain:** The performance of this stage is demonstrated in Figure 2c and Figure 3f bar plots from our simulation studies. These plots show the true positive and false positive rates for correctly detecting cell types within each domain. A high true positive rate in our results indicates effective identification of the correct cell types.
- 3) **Cell Type Proportion Estimation:** The performance evaluation for this stage is presented as scatter plots, scatter-pie plots, and boxplots in Figure 2, as well as in Supplementary Figures 5 and 6. These visual representations compare our method with other methods in our simulation study, providing a clear insight into the efficacy of our estimation process.

Since the output of a previous step serves as the input for the next stage, the comparison between the final estimated and true proportions also serves as an indicator of combined overall performance. Figures 2d, e, f, and Figure 3b, c, d, e, g collectively demonstrates the superior performance of SPADE.

4. While you discussed results and performance on simulated data, inclusion of results on real data would be beneficial for readers.

Thank you for highlighting the importance of applying our method to real data. We have indeed evaluated the performance of our approach using real datasets from diverse sources, such as Developmental Chicken Heart, Human Breast Cancer, and Mouse Visual Cortex. These applications are detailed in our manuscript.

It is crucial to note that unlike simulated data, where ground truth is known, real datasets do not provide a benchmark for true cell type proportions. Therefore, a direct comparison between estimated and actual cell proportions is not feasible. However, we have employed an alternative approach to validate our results. We used existing literature to provide context and reference for our estimated cell type patterns. For instance, in the case of the Developmental Chicken Heart dataset, we analyzed the estimated cell type proportions in the context of different developmental stages. This comparison allowed us to assess the plausibility and relevance of our estimates, ensuring that they align with established biological understanding and expectations. To better visualize and compare the real data results, in the revised version we have added the results from each method using real datasets in the supplementary file. We believe that this approach offers substantial evidence of our method's applicability and effectiveness in real-world scenarios, complementing our findings from simulated data.

5. The robustness of SPADE in the face of noise would be of interest. Perhaps the method's performance could be discussed in relation to at least two datasets?

Regarding the robustness of SPADE in the face of noise, we conducted additional simulation studies using both the mouse olfactory bulb data and the mouse kidney data. To assess the method's performance, we introduced different levels of noise into the simulated datasets. We then evaluated each method's performance using metrics such as mean Absolute Deviation (mAD), Root Mean Square Error (RMSE), and Pearson's Correlation Coefficient (R).

We have included line graphs in our revised supplementary file to visualize these comparisons for each dataset. In these graphs, each panel represents a designated metric, and the performance of each deconvolution method is indicated by colored lines. Our observations reveal that SPADE consistently yielded the low deviation in terms of mAD and RMSE. Moreover, SPADE has a high correlation in terms of R. Notably, SPADE's performance remained quite stable even as the noise level increased, indicating the superior robustness and effectiveness of our method.

Supplementary Figure 6. Performance comparison at different noise levels in the MOB data. The x-axis represents noise levels, and the y-axis is the values corresponding to each evaluation metric. Each panel illustrates a distinct evaluation outcome. Results obtained from different methods are represented by colored lines. Lower values are desirable for metrics like mAD and RMSD, while a higher value is preferable for the correlation coefficient, denoted as R.

Supplementary Figure 8. Performance comparison at different noise levels in the mouse kidney data.

The x-axis represents noise levels, and the y-axis is the values corresponding to each evaluation metric. Each panel illustrates a distinct evaluation outcome. Results obtained from different methods are represented by colored lines. Lower values are desirable for metrics like mAD and RMSD, while a higher value is preferable for the correlation coefficient, denoted as R.

We have also added extra contents to refer to the additional evaluation with different noise levels in the main manuscript as follow (line 132 and line 155):

“We have also considered adding different levels of noise when generating the synthetic data, and compared the performance of SPADE with other methods on noisy data. From Supplementary Figure 6, the results indicate that SPADE not only performs well under noisy conditions but also maintains its superior performance among all the compared methods.”

“To evaluate the ability of SPADE in handling noise data, we introduced varying levels of noise during the creation of synthetic data and compared its performance with other methods. The results, as shown in Supplementary Figure 8, reveal that SPADE not only copes well with noisy conditions but also continues to maintain low deviance and high correlation among all compared methods.”

6. In Figure 4, you discuss the significance of changing cell proportions over time. Could you elaborate on the importance of these changes? Are they indicative of cell transition? If so, some analysis or evidence to support this would be beneficial.

Thank you for your inquiry regarding the dynamic cell proportions illustrated in Figure 4, particularly in the context of the developing heart. This figure demonstrates the dominance of various cell types during heart development. Initially, we observe a predominance of immature myocardial and fibroblast cells, which is indicative of the early stages of heart formation. As development progresses, there is a noticeable shift where cardiomyocytes become more dominant, and the proportion of fibroblasts decreases. This trend suggests a transition in cell types corresponding to the maturation of the heart.

Your suggestion to provide additional evidence supporting this observation is well-taken. While further wet lab experiments could indeed lend more support to these findings, such experiments are beyond the scope of this current paper and would require considerable time and resources. Our primary goal in this paper is to show the capability of our method in identifying and estimating cell type proportions.

We recognize the importance of further validating these observations and are planning to incorporate additional in-depth analysis and experimental validation in an ongoing project. This future work aims to provide a comprehensive understanding of the biological significance and implications of these cell transitions during heart development.

7. Introduction and explanation of "colocalization analysis" and its importance would be beneficial.

Thank you for pointing out the need for a more detailed introduction and explanation of "colocalization analysis" in our manuscript. Cellular colocalization analysis is a critical tool in spatial transcriptomics, allowing us to investigate the relationships between different cell types within a tissue. In our study, we utilized this analysis to determine the correlation between distinct cell types.

The primary objective of this analysis is to evaluate spatial colocalization patterns, which involves assessing how closely different cell types are located to one another within the tissue. This proximity can give valuable insights into potential physical interactions and functional relationships between cell types. For example, cells that are frequently found in close proximity might be involved in similar or complementary biological processes or might interact with each other within the tissue's microenvironment.

Understanding these spatial relationships is essential in fields such as developmental biology, pathology, and tissue engineering, where the physical arrangement of cells can have significant implications for tissue function and disease progression.

In the revised manuscript, we have added the following sentences in the application on real data section (line 224) to provide readers with a clearer understanding of its role and importance in our study.

“We utilized cellular colocalization analysis, a key technique in spatial transcriptomics, to quantitatively evaluate how different cell types are positioned and interact within tissue. This approach provides insights into the spatial dynamics of cellular environments, revealing potential interactions and functional relationships between cells. By analyzing the spatial organization and proximity of cell types, we aim to understand their roles in tissue function and development, and how they contribute to the overall tissue architecture and intercellular communication.”

8. Improvement could be made to all sections, specifically the Abstract and Results sections, to enhance reader comprehension.

We have revised the Abstract, Introduction and Results section as below and marked all changes in red in the manuscript:

For abstract

“Understanding gene expression in different cell types within their spatial context is a key goal in genomics research. SPADE (SPAtial DEconvolution), our newly developed in silico method, addresses this by integrating spatial patterns into the analysis of cell type composition. This approach uses a combination of single-cell RNA sequencing, spatial transcriptomics, and histological data to accurately estimate the proportions of cell types in various locations. Our analyses of synthetic data have demonstrated SPADE's capability to discern cell type-specific spatial patterns effectively. When applied to real life datasets, SPADE provides new insights into cellular dynamics and the composition of tumor tissues. This enhances our comprehension of

complex biological systems and aids in exploring cellular diversity. SPADE represents a significant advancement in deciphering spatial gene expression patterns, offering a powerful tool for the detailed investigation of cell types in spatial transcriptomics."

Modified the beginning of 3rd paragraph in Introduction as below (line 32):

"Single-cell RNA sequencing (scRNA-seq) has significantly advanced our understanding of cell heterogeneity and gene expression patterns at an individual cell level [14]. While scRNA-seq reveals intricate details of cellular functions, its limitation lies in not capturing the spatial context of cells within tissues [9]. Addressing this gap, computational deconvolution techniques have emerged, focusing particularly on integrating spatial transcriptomics with single-cell data. This integration is vital for understanding tissue architecture and the spatial distribution of cell types."

Modified sentences for result sections:

The beginning of 2nd paragraph for the Developmental Chicken Heart section (line 167)

"During early embryonic development, the heart initially forms as a simple tube and undergoes a series of intricate morphological changes, eventually developing into a fully functional four-chambered heart complete with the blood vessels. Mantri, M. et al. employed a combination of spatially resolved RNA sequencing and high-throughput single-cell RNA sequencing to investigate the spatial and temporal interactions as well as the regulatory mechanisms involved in the development of the embryonic chicken heart [26]."

The second half of the 1st paragraph for the Human Breast Cancer Data section (line 245)

"The combination of spatial transcriptomics and single-cell data is proving to be a valuable method for unraveling the complexities of human breast cancer [52]. This method maps gene expression and analyzes single-cell transcriptomes to identify cell types and their interactions in the tumor environment, crucial for understanding cancer progression and treatment effectiveness. "

The 1st and 2nd paragraphs for the Mouse Visual Cortex Data section (line 282)

"The mouse brain, with its millions of neurons, is an ideal model for studying mammalian brain structure and function, especially in the visual cortex. This region, crucial for processing visual information, is organized into layers, each with specialized cell types, making it a good model for human visual cognition research [60, 61, 62, 63]. The visual cortex hosts various neuron types, including excitatory neurons using glutamate and inhibitory neurons using GABA, forming a network for interpreting visual stimuli [64, 65]. Each neuron type plays a specific role in visual processing, from detecting visual features to integrating complex visual information [66, 67, 68]. Understanding these functions and their disruptions can provide insights into neurological and psychiatric disorders [69].

We implemented a single-cell analysis [59] to identify 30 distinct cell types in the mouse visual cortex. This analysis was used to deconstruct the adult mouse brain, which had undergone

spatial processing (see Figure 6a). Initially, we divided the mouse brain into 19 different regions (illustrated in Figure 6b). In these regions, we were able to identify specific layers that correlate with various brain functions. Using SPADE, we successfully decomposed each brain region into its constituent cell types. The predominant cell type in each location is shown in Figure 6c. Our focus was particularly on the visual cortex, where we found that most areas were primarily composed of excitatory neurons, followed by inhibitory neurons and oligodendrocytes, as detailed in Figure 6d."

Discussion section (line 307):

"Spatial transcriptomics, essential for studying gene expression and tissue diversity, is more informative when combined with cell type deconvolution. This computational method identifies cell types from gene expression data, enhancing our understanding of biological processes at the cellular level within tissues."

The Discussion section has also been modified to incorporate the comments from reviewer 2.

Your research is indeed compelling, and with these additional details, it promises to be even more insightful and accessible to readers.

Reviewer #2 (Remarks to the Author):

Overall the concept and the work seemed both important and nice to me. And this will be very important to reveal the disease biology of different tissue. The writing language is very lucid, easy to understand and conceptualize.

Major comments:

1. In the stage of applying the method in real-world data, it is unclear if histology data is required. Within the "Application on Real Data" section the authors used histology data for the first and the third experiment, but didn't mention histology data on the second experiment where they applied SPADE to the breast cancer dataset.

We used histology data as an input for the human breast cancer data, same as for all other experiments. However, the corresponding histology image was not included in the main text of the manuscript in our first submission. During revision, we have added the histology image to the supplementary figure 16, shown below.

Supplementary Figure 16. Human breast cancer. Left is the H&E staining for breast cancer tissue. Right is the scatter pie plot for estimated cell types at each location. Each location is represented by a pie plot colored by different cell type composition. The cancer epithelial cells are dominated at most of locations, followed by plasmablasts and myeloid cells.

2. It will be helpful to know if this method can be applied to spatial datasets where instead of a whole transcriptome, a panel of genes were used.

Our method relies on gene expression data for cell type selection and proportion estimation. While it is possible to apply our method using only a panel of genes, this approach may influence the results. Since the genes are used across each step, especially to distinguish cell types, the ability of the panel of genes to successfully identify cell types would require further investigation. If the marker genes are not included in the panel, then it is likely to fail in identifying the cell type.

3. It is evident that single cell (reference) data set will not be from the same patient and the same region from where the spatial dataset will be collected. Thus it will be important to know how this tool will handle conditions where there are some domains containing some additional cell types or states, not identified in single cell (reference) dataset. This issue can be addressed by a simulation approach.

Addressing the discrepancy in cell types between single-cell reference datasets and spatial datasets is a central focus of our ongoing research. The challenge lies in identifying and quantifying missing cell types in the reference dataset. Our approach includes two main strategies: firstly, enriching the reference dataset with comprehensive data to encompass a broader range of cell types, and secondly, developing a new algorithm capable of predicting the

number of missing cell types. We are actively pursuing these directions and hope to present our findings in a future study.

4. How does the presence of rare cell types affect deconvolution?

Indeed, accurately estimating the proportions of rare cell types remains a challenging aspect for most deconvolution methods. Our observations from simulations using mouse olfactory bulb data, particularly with the Immature cells (as shown in Figure 2d), indicate that our method, similar to most other methods, tends to underestimate rare cell types. We acknowledge this as a limitation of our current approach. To address this, we have added a discussion in our manuscript about this limitation and outlining our ongoing efforts to enhance the method's accuracy for rare cell types in future work.

We have modified the discussion section to improve discussion regarding rare cell type (line 330):

“Although the SPADE algorithm has demonstrated superior accuracy, one of its notable challenges is the accurate deconvolution of rare cell types. Our analysis, especially with rare cell types like the Immature cells in the mouse olfactory bulb data, indicated a tendency for underestimation, a limitation common to current deconvolution methods. This underestimation issue is critical to address in order to improve SPADE's robustness and applicability, particularly in complex biological tissues where rare cell types are crucial for functional significance or disease state.

Moreover, it is important to note that SPADE's performance can be further improved by incorporating better-designed reference datasets. In our study, we utilized one single scRNA-seq dataset to construct the reference, potentially limiting the algorithm's overall efficacy. Advances in scRNA-seq technologies have led to the generation of multiple reference datasets from different platforms or samples obtained from the same tissues. Integration of these diverse scRNA-seq datasets holds the potential to provide a comprehensive and accurate reference set, thereby improving the performance of the SPADE algorithm.

.....

Continuous methodological improvements are necessary. Future studies should explore the use of multiple reference datasets to improve the accuracy and efficacy of SPADE in predicting cell types and their spatial distribution across different tissues.”

5. When simulating data for cell type selection, how is care taken to avoid bias which might result from cells with very extreme expression profiles or cells over-represented in the data?

For the selection of cell types when simulating synthetic data, we utilized all cell types present in the single-cell data to construct this synthetic dataset. The detailed steps for this process are explained in Supplementary Figure 2.

To determining cell types for each domain, which is the second step of our method, we used lasso regression. This approach integrates spatial gene expression with cell type-specific gene expression, aggregating individual cell expressions into cell type expressions. Such aggregation effectively minimizes the impact of extreme values from individual cells on our estimations. Furthermore, in constructing the cell type reference, a detailed explanation is provided in Supplementary Figure 3. Here, our method incorporates scaled cell counts in calculating the final cell type counts, thereby ensuring that extreme values do not distort the results. This methodology ensures a more balanced and representative analysis, mitigating the risk of bias due to cells with extreme expression profiles or overrepresentation.

6. The results from other deconvolution methods applied to real datasets need to be included and compared with the SPADE results.

In response to your suggestion, we have included results from cited deconvolution methods for the Developmental Chicken Heart, Human Breast Cancer, and Mouse Visual Cortex datasets. These are now included in the supplementary files as Supplementary Figure 11 -15 for the Developmental Chicken Heart data, Supplementary Figure 17 for the human breast cancer data, and Supplementary Figure 18 for the mouse visual cortex data. Given the absence of ground truth for these real datasets, we are unable to employ evaluation metrics such as RMSE for performance comparison. However, we have provided a comparative analysis based on the overall trends observed for each cell type within the respective datasets.

The additional figures for the results of real data estimated by cited methods are as follow:

Supplementary Figure 11. Estimated cell type proportion for the Chicken heart data from CARD. colors representing different cell types.

Supplementary Figure 12. Estimated cell type proportion for the Chicken heart data from RCTD. colors representing different cell types.

Supplementary Figure 13. Estimated cell type proportion for the Chicken heart data from SPOTlight. colors representing different cell types.

Supplementary Figure 14. Estimated cell type proportion for the Chicken heart data from SpatialDecon. colors representing different cell types.

Supplementary Figure 15. Estimated cell type proportion for the Chicken heart data from spatialDWLS. colors representing different cell types.

Supplementary Figure 17. Comparison of the estimated cell type proportion for the human breast cancer data. Colors representing different cell types. Each bar represents the proportion estimated from difference methods.

Supplementary Figure 18. Comparison of the estimated cell type proportion for the mouse visual cortex data. Colors representing different cell types. Each bar represents the proportion estimated from difference methods.

7. How does SPADE take care of spatial noise and batch effects that might be present in spatial transcriptomics data?

For spatial noise: we performed additional simulation study, and the simulated datasets contain 4 levels of noise. The results are shown in the response to comment 5 from the reviewer #1.

For spatial batch effect: the current version of our method focuses on cell type deconvolution and was tested on data that were preprocessed to minimize batch effects. Therefore, the current version does not specifically address these effects. We recognize the significance of batch effects, as detailed in "Computational Approaches and Challenges in Spatial Transcriptomics." [1]. This complexity, including various types of batch effects, is a known challenge in the field. Our future work will explore the integration of batch effect correction methods into our algorithm. This enhancement will be aimed at improving the method's robustness and applicability to a broader range of datasets, addressing a critical aspect of spatial transcriptomics analysis.

Minor Comments:

Introduction:

1. The introduction part can be made little brief by minimizing statements on

scRNA-seq and deconvolution of scRNA-seq and more focusing on spatial deconvolution.

We have merged two paragraphs into one. This revised section presents the necessary background with minimal statements on scRNA-seq and bulk deconvolution. The updated paragraph is as follows (line 32):

“Single-cell RNA sequencing (scRNA-seq) has significantly advanced our understanding of cell heterogeneity and gene expression patterns at an individual cell level [14]. While scRNA-seq reveals intricate details of cellular functions, its limitation lies in not capturing the spatial context of cells within tissues [9]. Addressing this gap, computational deconvolution techniques have emerged, focusing particularly on integrating spatial transcriptomics with single-cell data. This integration is vital for understanding tissue architecture and the spatial distribution of cell types. ”

2. It would be better if one statement is there about time and memory effectiveness of SPADE over other spatial deconvolution methods available under comparable system configurations. That would be more attractive to the users!

In terms of memory usage, SPADE is designed with moderate memory efficiency in mind, balancing the computational demands with the method's performance capabilities. This ensures that SPADE is accessible for use in a range of system configurations without excessive memory requirements. As for time efficiency, SPADE does not lead among the six evaluated methods, primarily due to an additional, crucial step in our approach that involves identifying cell types within each domain. While this step contributes to a longer processing time, it is essential for enhancing the accuracy and robustness of SPADE, especially when dealing with complex datasets. This is similar to spatialDWLS, the only other method in our comparison that incorporates a comparable cell type selection step, which also results in longer processing times due to this added complexity.

We have updated the discussion section (line 344) in our manuscript to highlight this trade-off, emphasizing how SPADE's thorough analysis compensates for its moderate efficiency.

“It should also be noticed that SPADE, while not the most efficient in processing time and memory usage, involves a process of identifying cell types within each domain before estimating proportions. This step, though extending processing time, substantially enhances the accuracy and robustness of our analyses, particularly for complex datasets. This balance between processing efficiency and analytical precision is a key consideration, making SPADE a valuable tool for in-depth spatial gene expression studies.”

3. I am really eager to understand how SPADE will work if histology images are not available for some spatial datasets? How it will identify spatial domains in these cases?

SPADE uses spaGCN, an existing method, for detecting spatial domains. One of the key strengths of spaGCN is its ability to leverage image data to enhance the cohesion of spatial

location analysis. Significantly, this method is also adaptable to datasets lacking histology images. In such cases, spaGCN utilizes spatial gene expression data to identify spatial domains, a process similar to other methods focusing on spatial domain detection. While our paper does not delve into the detailed algorithm of spaGCN, this aspect is thoroughly covered in the original publication, offering a comprehensive understanding of its application in diverse data scenarios.

We have added this explanation to the Method section (line 367):

“Importantly, spaGCN can be applied to the datasets where histology images are absent. In these situations, it makes use of spatial gene expression data to identify the spatial domains which is comparable to methods used in other spatial domain detection approaches.”

Results:

1. Figure - 1: The layout is completely fine. Just a technical error is that the ‘L’ of locA.3 in Domain A of the figure is in lower case while all other is in upper case.

We have modified the figure using all upper case.

2. Figure - 2: In text explanation (mouse olfactory bulb), it would be better if one statement is added with respect to significance of SPADE having least mAD, RMSE and highest R across simulations.

We have added the following sentence to the figure 2 legend.

“The overall simulation results indicate that SPADE outperformed other methods, achieving the lowest mean Absolute Deviation (mAD), the least Root Mean Square Error (RMSE), and the highest R.”

Discussion:

1. It should include all strengths of SPADE for example along with spatial domain, any other advantages over other methods.

Thank you for your valuable feedback. Our method, SPADE, not only integrates spatial domain structures but also uniquely excels in selecting cell types for each domain. This feature distinguishes SPADE from most existing cell type deconvolution methods. To our knowledge, spatialDWLS is the only other method that implements cell type selection prior to estimating proportions. However, spatialDWLS relies on enrichment tests followed by a global cutoff for cell type selection. This approach is not only arbitrary and unstable but also time-consuming. In contrast, SPADE employs a lasso regression combined with an adaptive thresholding technique, offering a more data-driven and flexible solution for accurately detecting the correct cell types. This advantage is clearly demonstrated in Figure 2C, where the bar plot highlights SPADE's superior performance in cell type selection compared to other methods.

We have added the following to the Discussion section (line 319) to emphasize our strength:

“Our method stands out by integrating spatial structures and using a novel approach for cell type selection. Differing from other techniques, it employs lasso regression and adaptive thresholding for more accurate and flexible cell type identification. This effectiveness is evident in our results, notably in Figure 2C, enhancing SPADE's robustness and precision in complex spatial transcriptomics datasets.”

2. Also if there are some significant differences between synthetic or simulated datasets vs. real-world datasets on the performance of SPADE.

Our synthetic datasets were carefully constructed to closely mirror real-world data, with detailed steps outlined in Supplementary Figure 2. Briefly, the process involved selecting cells from single-cell datasets to represent various cell types. These cell types were then mapped to specific locations to generate the count for each location. The location for each domain is determined based on the real spatial data. However, it's important to note that the proportions of cell types at each location were randomly generated by the Dirichlet distribution which is widely used in this area. This approach is used widely in simulating data for assessing deconvolution methods. By using this method, we ensured that the synthetic data provided a realistic base for evaluating SPADE's effectiveness, allowing us to draw meaningful comparisons between its performance on synthetic and actual datasets.

[1] S. Fang *et al.*, “Computational Approaches and Challenges in Spatial Transcriptomics,” *Genomics Proteomics Bioinformatics*, vol. 21, no. 1, pp. 24–47, Feb. 2023, doi: 10.1016/j.gpb.2022.10.001.